# α-Gal immunization positively impacts *Trypanosoma cruzi* colonization of heart tissue in a mouse model

**Gisele Macêdo Rodrigues da Cunha**[1☺], **Maíra Araújo Azevedo**[1☺], **Denise Silva Nogueira**[1], **Marianna de Carvalho Clímaco**[1], **Edward Valencia Ayala**[2], **Juan Atilio Jimenez Chunga**[3], **Raul Jesus Ynocente La Valle**[3], **Lucia Maria da Cunha Galvão**[1], **Egler Chiari**[1†], **Carlos Ramon Nascimento Brito**[4], **Rodrigo Pedro Soares**[5], **Paula Monalisa Nogueira**[5], **Ricardo Toshio Fujiwara**[1], **Ricardo Gazzinelli**[1,5], **Robert Hincapie**[6], **Carlos-Sanhueza Chaves**[6], **Fabricio Marcus Silva Oliveira**[1], **M. G. Finn**[6], **Alexandre Ferreira Marques**[1] *

1 Universidade Federal de Minas Gerais, Departamento de Parasitologia, Belo Horizonte, Brazil,
2 Universidad San Martin de Porres, Lima, Peru, 3 Universidad Nacional Mayor de San Marcos, Faculdad de Ciencias Biologicas, Escuela Profesional de Microbiología y Parasitología—Laboratorio de Parasitología en Fauna Silvestre y Zoonosis, Lima, Peru, 4 Universidade Federal do Rio Grande do Norte—Centro de Ciências da Saúde—Departamento de Análises Clínicas e Toxicológicas, Natal, Brazil, 5 Instituto René Rachou/FIOCRUZ–MG, Belo Horizonte, Brazil, 6 School of Chemistry and Biochemistry, School of Biological Sciences, Georgia Institute of Technology, Atlanta, Georgia, United States of America

☺ These authors contributed equally to this work.
† Deceased.
* amarques8@gatech.edu

**Data Availability Statement:** All relevant data are within the manuscript.

## Abstract

Chagas disease, caused by the parasite *Trypanosoma cruzi*, is considered endemic in more than 20 countries but lacks both an approved vaccine and limited treatment for its chronic stage. Chronic infection is most harmful to human health because of long-term parasitic infection of the heart. Here we show that immunization with a virus-like particle vaccine displaying a high density of the immunogenic α-Gal trisaccharide (Qβ-αGal) induced several beneficial effects concerning acute and chronic *T. cruzi* infection in α1,3-galactosyltransferase knockout mice. Approximately 60% of these animals were protected from initial infection with high parasite loads. Vaccinated animals also produced high anti-αGal IgG antibody titers, improved IFN-γ and IL-12 cytokine production, and controlled parasitemia in the acute phase at 8 days post-infection (dpi) for the Y strain and 22 dpi for the Colombian strain. In the chronic stage of infection (36 and 190 dpi, respectively), all of the vaccinated group survived, showing significantly decreased heart inflammation and clearance of amastigote nests from the heart tissue.

## Author summary

Chagas disease, caused by the protozoan parasite *Trypanosoma cruzi*, is a significant endemic infectious disease in Latin America and is spreading in the U.S. and Europe with the presence of its insect transmission vector. No approved vaccine against Chagas disease

**Funding:** This work was supported by the Conselho Nacional de Desenvolvimento Cientifíco e Tecnoloǵico, CNPQ Brazil (Projects 407926/2018-6; 303698/2019-5) to AFM, and a research partnership between Children's Healthcare of Atlanta and the Georgia Institute of Technology. National Institute of Health (1R01AI116577) to MGF; National Institute of Science and Technology for Vaccines/Conselho Nacional de Desenvolvimento Científico e Tecnológico (465293/2014-0) to RTG. The funders had no role in study design, data collection, and analysis, decision to publish, or preparation of the manuscript.

**Competing interests:** The authors have declared that no competing interests exist. Author Professor Egler Chiari was unable to confirm their authorship contributions. On their behalf, the corresponding author has reported their contributions to the best of their knowledge.

exists. We describe a vaccine candidate based on a carbohydrate found on the *T. cruzi* cell surface, linked in the vaccine to a virus-like particle that provides a strong and focused immune response. Mice were immunized and challenged with the *Trypanosoma cruzi* parasites from two strains (Y and Colombian). Vaccination conferred substantial protection of mice against infection, compared with the unvaccinated group. Vaccinated animals presented low parasitemia, increased production of pro-inflammatory cytokines IL-12 and IFN-γ, decreased cardiomyocyte damage, and rapid clearance of parasite nests from heart tissue. These effects were especially significant at time points modeling chronic disease, an important consideration for this pathogen. We, therefore, believe this is a valuable path to pursue in the development of vaccines against Chagas disease.

## Introduction

Chagas disease is caused by the protozoan parasite *Trypanosoma cruzi*, commonly transmitted through the feces of the infected reduviid bug (kissing bug). The disease cycle was discovered in 1909 by the Brazilian physician Carlos Chagas [1]. It is estimated that 7 million people are infected with *T. cruzi*, with approximately 18,000 new cases each year, [2] representing a serious health problem in Latin America that is spreading in the U.S. [3] and globally [4]. The acute stage of the disease can have symptoms similar to viral infection or febrile illness, but infected patients may progress to acute myocarditis, and meningoencephalitis may occur [5]. While patients with chronic Chagas disease can exhibit no adverse heart disease complications, dilated cardiomyopathy with heart failure is the most common cause of death among these patients [6–8]. Treatment is limited to the two drugs available, nifurtimox and benznidazole. Their efficacy in the acute phase of the disease is questionable, [9] costly, and significant side effects can present as gastrointestinal distress; cutaneous hypersensitivity and neurological symptoms have also been reported.[10] With no vaccine currently available for Chagas disease, [11,12] a new immunological approach is needed [13,11,14]. An extensive range of vaccine formulations has been assessed in recent years, from the use of whole attenuated parasites to purified or recombinant proteins, viral vectors, and DNA vaccines [15–19].

Because humans lack the galactosyltransferase activity necessary to construct it, antibodies against the α-Gal (Gal-α1,3-Gal-β1,4-GlcNAc) motif are the most abundant natural antibody in humans [20–22]. The α-Gal trisaccharide has therefore attracted attention as a molecular adjuvant in vaccines against a variety of targets, including cancer [23,24] and wound healing applications [25–27]. It also represents an important antigen in its own right, as α-Gal is a cell-surface marker of *Leishmania*, [28,29] malaria, [30] and *T. cruzi* [31]. Importantly, in the latter case, both CD4 and CD8 T cell activation was reported.

The early identification and response to *T. cruzi* are mediated mainly by the TLR family of type I transmembrane receptors. The surface of *T. cruzi* contains large amounts of glycoinositolphospholipids (GIPLs), presented alone or as anchors for glycoproteins and polysaccharides.[32,33]. Macrophages respond to these ligands by producing proinflammatory cytokines, which are crucial to controlling *T. cruzi* infection and disease outcome [34,35]. TLR4 expression has been reported to be particularly important in this regard [36]. As previously described with *Leishmania*, [28] we report here that vaccination of α-galactosyltransferase knockout (αGalT-KO) mice using VLPs displaying the α-Gal epitope produced higher titers of anti-αGal IgG antibodies and protection against infection by *T. cruzi* Y and Colombian strains, which display different levels of galactosylation in their glycoconjugates. Particularly notable

was the effect of such immunization on the control of heart inflammation and the clearance of parasite nests from heart tissue.

## Results

We have previously described the preparation of Qβ virus-like particles functionalized with a dense array of synthetic α-Gal trisaccharide (designated Qβ-αGal, with 540±50 trisaccharides per particle) and the use of those particles to elicit high titers of anti-αGal IgG antibodies in αGalT-KO mice [28,37]. These particles are approximately 30 nm in diameter, highly homogeneous in size, with most of the surface lysine residues acylated with a 10-atom linker bearing a terminal azide group. To these azide groups were conjugated αGal bearing a β-linked short chain with a terminal alkyne group. The rapid and selective nature of the copper-mediated azide-alkyne ligation reaction [38] is required to achieve this high-density display of the αGal motif.

To test protection against *T. cruzi* injection, αGalT-KO mice received a 10-μg dose of Qβ-αGal, with control groups receiving unfunctionalized Qβ VLPs or PBS (**Fig 1A**). The vaccinated group showed significantly higher IgG anti-αGal antibody titers (**Fig 1B**) after one week. A boost injection of the same dose was followed one week later with an intense challenge of $10^6$ Y strain (TcII) parasites (recovered previously from mice in a standard passaging procedure). Parasitemia was controlled much better in the vaccinated group (**Fig 1C**), and 58% of this group survived more than 50 days compared to complete mortality within two weeks post-infection in the control groups (**Fig 1D**).

To further assess Qβ-αGal as a candidate for vaccine development against *T. cruzi* infection, the effectiveness of a single immunization was tested against a more typical lower parasite dose of the Y strain. Experimental groups (6–10 mice) were sacrificed at either 8 or 36 dpi, modeling acute and chronic of Chagas disease stages, respectively. Given the high anti-αGal responses (**Fig 1B**), we challenged immunized αGal-KO mice with $10^4$ parasites each one week after a single immunization. Parasitemia (**Fig 2A**) and survival rates (**Fig 2B**) were again

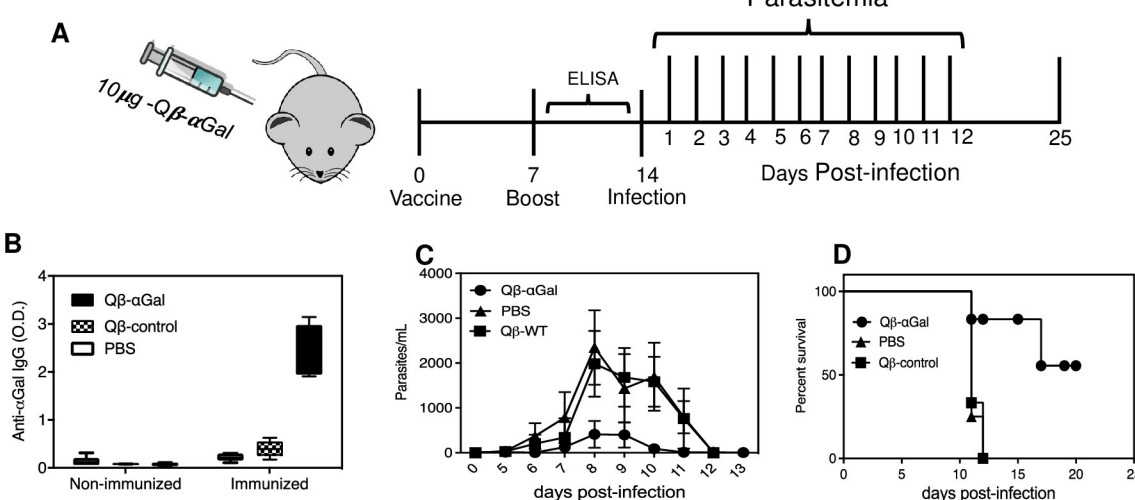

**Fig 1. Immunization against high dose of *T. cruzi* Y strain.** (A) Immunization and analysis schedule. αGalT-KO mice (5 per group) were inoculated (subcutaneous) with two 10 μg doses of of Qβ-αGal, Qβ-WT, or PBS on days 0 and 7. On day 14 (dpi 0), all groups were infected with $10^6$ Y strain. (B) Relative anti-α-Gal IgG serum antibody levels measured by ELISA before infection, Y days before infection (serum dilution = 1/100). (C) Parasite levels in blood checked daily starting at day 1 post-infection (dpi). (D) Mouse survival. All experiments were performed independently in triplicate.

dramatically improved; because of the lower parasite dose, approximately 60% of the control animals survived, compared to 100% of the immunized groups. Since heart damage is a hallmark of *T. cruzi* parasite infection, heart tissue from all animals was examined and scored for signs of degeneration, inflammation, cardiomyocyte hypertrophy, and parasite lesions [39–41]. While degenerative damage was observed in all animals in the acute phase, this was completely resolved for the immunized animals, in contrast to the control group (**Fig 2C**). Similarly, abnormal cardiomyocyte hypertrophy was enhanced in the control group but missing among the immunized group (**Fig 2D**). Inflammation was suppressed by immunization at both 8 and 36 dpi (**Fig 2E**), and very few parasites were found in the heart at day 36 (**Fig 2F**).

The myotropic Colombian (TcI) strain of *T. cruzi* is associated with Chagas disease cardiomyopathy [42–44]. Groups of 7–10 αGalT-KO mice were vaccinated and infected with the same protocol as above using a standard challenge of $10^3$ parasites per mouse. Parasitemia was followed for 45 days, and all groups of mice were kept for more than 190 days. All vaccinated αGalT-KO mice showed higher anti-αGal IgG antibody levels throughout the experiment (**Fig 3A**). Parasitemia was well controlled in the vaccinated group (**Fig 3B**). Analysis of heart sections samples from all groups at 22 and 190 dpi showed the vaccinated αGalT-KO mice to have significantly fewer degenerative changes at 190 dpi (**Fig 3C**) and lower inflammation at both acute and chronic stages (22 and 190 dpi, respectively, **Fig 3E**). We observed a significant decrease of cardiomyocyte hypertrophy of vaccinated animals at 22 dpi; by 190 dpi, hypertrophy had largely resolved for both groups, but was undetectable in the Qβ-αGal-immunized mice (**Fig 3D**). Lesion intensity significantly decreased in the myocardium of the vaccinated group at 22 and 190 dpi compared with the animals receiving the underivatized particle (**Fig 3F**).

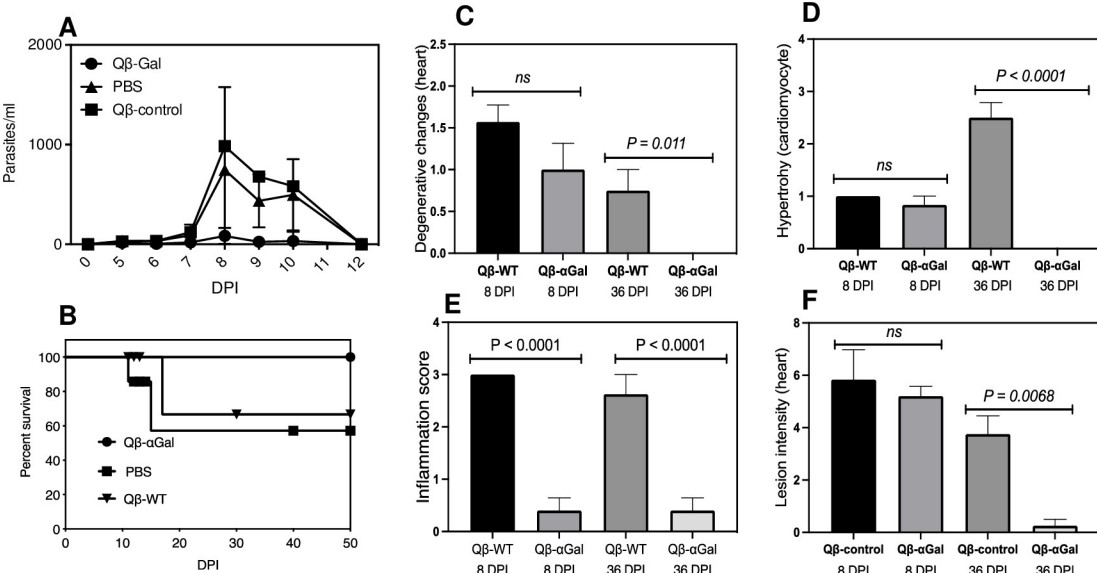

**Fig 2. Immunization against moderate dose of *T. cruzi* Y strain.** αGalT-KO mice (6–10 per group) were inoculated (subcutaneous) with one 10 μg dose of Qβ-αGal, Qβ-WT, or Qβ-Glu (alternate control bearing glucose units instead of α-Gal). Mice were challenged one week after immunization with $10^4$ Y strain. (A) Parasite levels in blood checked daily starting at day 1 post-infection (dpi). Red dashed lines show the upper limit of the standard deviation for each set of data. (B) Mouse survival. (C) Degenerative changes in the heart. (D) Hypertrophy. (E) Inflammation score. (F) Lesion intensity. All experiments were performed independently in triplicate. Significance p < 0.05. ns (not significant). Data plotted as SEM (Standard Error of the Mean).

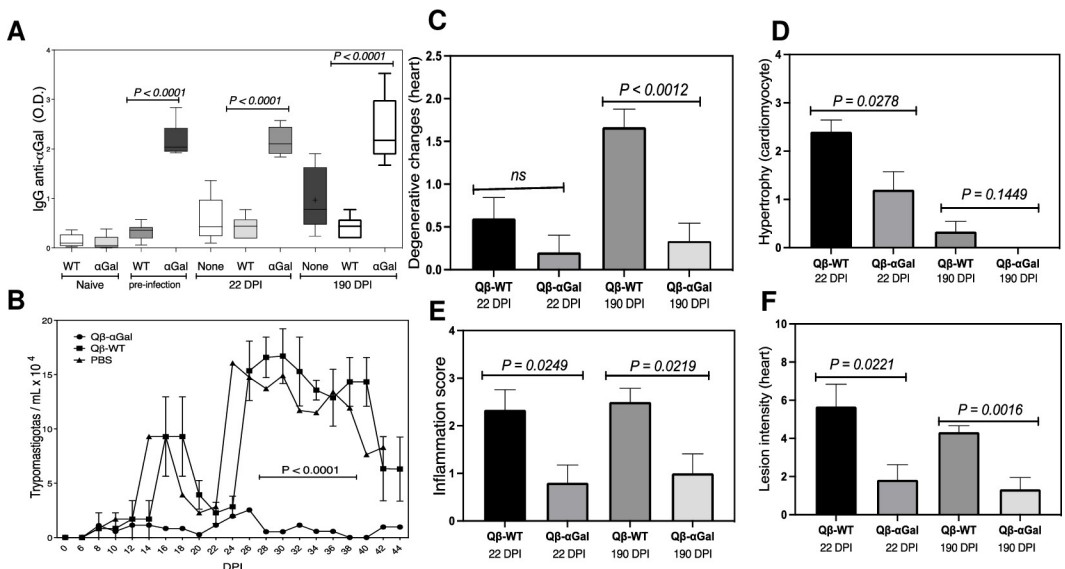

**Fig 3. Immunization against *T. cruzi* Colombian strain.** αGalT-KO mice (6–10 per group) were inoculated (subcutaneous) with two 10 μg doses of Qβ-αGal or Qβ-WT on days 0 and 7. On day 14, all groups were challenged with $10^3$ parasites of the Colombian strain. (A) Relative anti-α-Gal IgG serum antibody levels measured by ELISA (serum dilution = 1/100). "Pre-infection" denotes day 7, one week before challenge. (B) Parasite levels in blood checked daily starting at day 2 post-infection (dpi). Red dotted lines show the upper limit of the standard deviation for each set of data. (C) Degenerative changes in the heart. (D) Hypertrophy. (E) Inflammation score. (F) Lesion intensity. All experiments were performed independently in triplicate. Significance p < 0.05. ns (not significant). Data plotted as SEM (Standard Error of the Mean).

TLR4 has been reported to play an essential role in protecting mice against experimental *T. cruzi* infection, [36,45] and a synthetic TLR-4 agonist was found to increase mice survival and decrease cardiac colonization by the parasite [46]. We therefore tested the TLR agonistic potential of Qβ-αGal particles in this context using standard C57BL/6 mice, as well as their TLR knockouts, because the corresponding TLR variants of the GalT-KO strain were not available. We believe these results to be potentially relevant because there is some IgG anti-α-Gal immune response in C57BL/6 mice [47]. We collected peritoneal macrophages from unvaccinated wild-type, TLR 2[-/-], and TLR 4[-/-] knockout mice and treated them with unfunctionalized and α-Gal decorated VLPs. Qβ-αGal was uniquely found to induce significant nitric oxide production (although not to the level of lipopolysaccharide positive control) in wild-type and TLR2-knockout, but not in TLR4-knockout mice (**Fig 4A**), suggesting its action as a TLR4 agonist. Next, we evaluate cytokine production on heart homogenates. We observed higher concentrations of IL-12 and IFN-γ in homogenates from heart tissue of vaccinated and challenged groups of αGalT-KO mice at dpi 8 and 36 for Y strain and dpi 20 and 190 for Colombian *T. cruzi* infection (**Fig 4B and 4E**).

The presence of amastigote parasites was analyzed in 100 fields per slide of heart sections from vaccinated and unvaccinated αGalT-KO mice. Although parasites were found in the heart of the vaccinated group at 8 dpi for Y *T. cruzi* infection, they were cleared by day 36 (**Fig 4F**). Analysis at both dpi 22 and 190 after Colombian strain infection showed no amastigote nests for the vaccinated animals (**Fig 4G**). Representative histopathology images are shown in **Fig 4H**.

## Discussion

In Latin America, infection with the *Trypanosoma cruzi* parasite is the most common cause of inflammatory heart disease [48]. Chagas heart disease is characterized by acute and chronic stages, which can occur decades apart. The disease's effects can lead to a spectrum of problems, from organ damage due to high parasitemia in the acute stage (the first 2–3 months) to cardiac

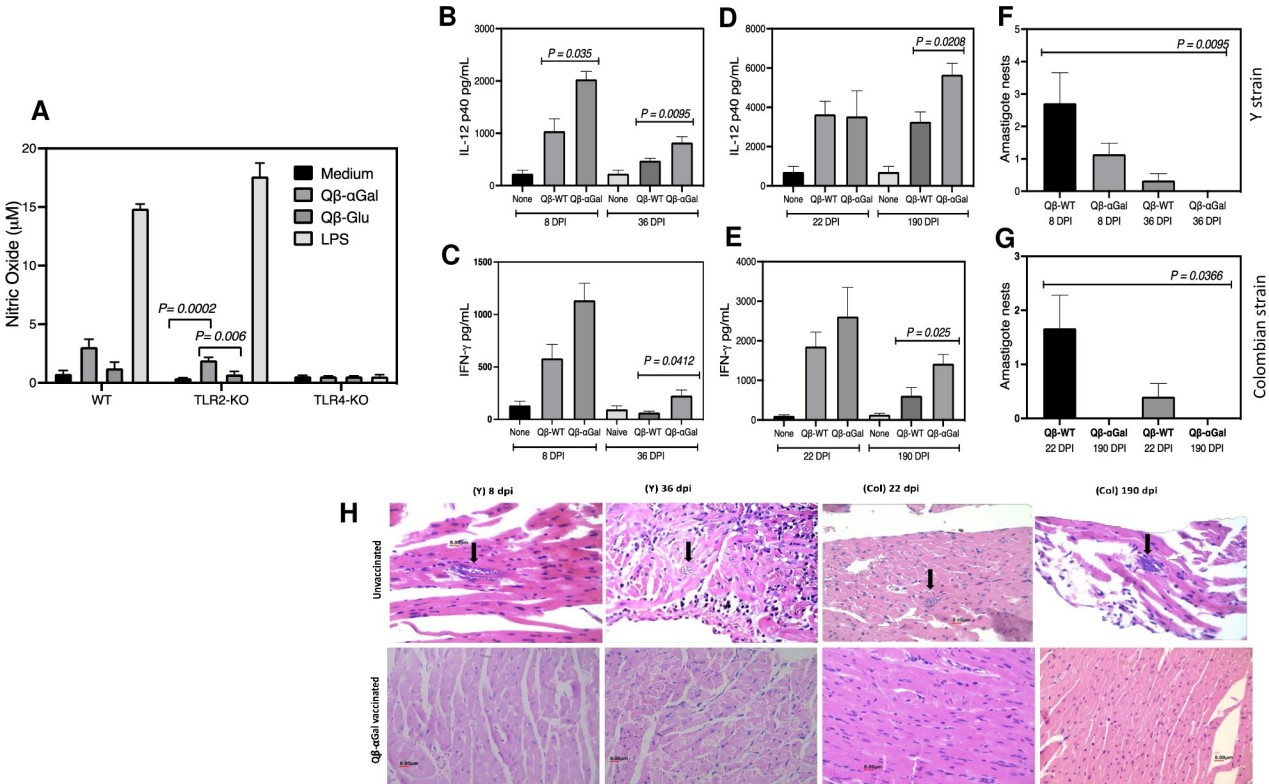

**Fig 4. TLR activity, cytokine production and histopathology.** (A) Nitric oxide production from stimulated peritoneal macrophages from unvaccinated mice of the indicated strains. Treatment groups refer to 1 μg/mL concentrations of the indicated agent; "medium" = buffer. (B and C) IL-12 and IFN-γ production by heart homogenate from vaccinated or unvaccinated αGalT-KO mice infected with Y strain. (D and E) IL-12 and IFN-γ production by heart homogenate from vaccinated or unvaccinated αGalT-KO mice infected with Colombian strain. (F and G) Average number of *T. cruzi* amastigote nests found in heart tissue by counting in 100 fields per slide of heart section. (H) Representative histopathological sections from the indicated mice; for example, the top-left panel shows heart tissue from an unvaccinated animal 8 days after infection with *T. cruzi* Y strain. Arrows point to amastigote colonies. All experiments were performed independently in triplicate.

lesions and neurogenic disturbances in the chronic stage [49,50]. Although 60% of infected patients can survive for decades with the disease, about 30–40% of patients develop heart manifestations in the chronic stage that may lead to death.[51–53] It is suggested that the persistence of amastigote parasites in heart tissues correlates directly with the disease severity due to direct disruption of host cells by parasite multiplication and exacerbation of inflammation caused by parasite residue [54–58]. Furthermore, the recommended drug treatment for Chagas disease, benznidazole, is not very efficient and is highly toxic upon sustained use [59].

The search for a vaccine against *T. cruzi* started in 1912 when it was observed that animals surviving acute infection became resistant to a second infection.[60] Since then, a variety of experimental vaccine candidates have been tested, involving live, killed, or attenuated parasites, recombinant *T. cruzi* proteins, peptides, and DNA [11,61–67]. Both the acute and chronic stages of Chagas disease can be reflected in several experimental animal models, [68–70] most examples using the Colombian (TcI) and Y (TcII) strains of *T. cruzi* [71].

Chagas disease patients from endemic areas are reported to have high levels of antibodies against the α-Gal glycotope, which is expressed by *T. cruzi* amastigotes and trypomastigotes. These antibodies are thought to confer protection to these individuals in the acute and chronic phases of the disease [72–74]. Following the report from Almeida and colleagues [31], our group also validated the α-Gal transferase knockout (αGalT-KO) mouse as an effective experimental Chagas disease model that mimics the key immunological property of humans and

Old-World primates [22,23,75] of producing high titers of anti-α-Gal antibodies [76]. Having earlier found that the virus-like particle bearing a high density of α-Gal is an effective experimental vaccine candidate against *Leishmania* spp., [28] we describe here the application of this particle to *T. cruzi*. Indeed, Almeida's use of α-Gal linked to human serum albumin [31] gave very promising results: the vaccinated group of αGalT-KO mice were protected against *T. cruzi* infection and presented higher anti-αGal antibody titers [31]. To this important precedent we add several important factors.

First, this is the first work comparing two different *T. cruzi* strains belonging to different DTUs [77]. These remarkable genetic polymorphisms are significant in scope and may hinder vaccine efficacy. For example, the strains used here display very different levels of galactosylation in their GPI-mucins, the Y strain being high and the Colombian strain low in this parameter [78]. That the Qβ-αGal particle was effective for both further supports its consideration as a promising vaccine candidate. Activity against the Colombian strain suggests that are other α-galactosyl-containing GIPLs (or other cell-surface glycoconjugates) may be involved in the immune response to this pathogen. Second, the vaccine used here is far better defined than the "Galα3LN-HSA" material employed by Almeida and colleagues. Purchased from a commercial supplier, we assume that it is created by conjugation to HSA lysine groups by virtue of an activated ester variant of the α-Gal trisaccharide using a "3 atom spacer" as described by the supplier. No characterization data is available, so neither the number of carbohydrates nor the homogeneity of the material are known. Virus-like particle platforms are, by contrast, homogeneous in size, shape, and in the number and location of the lysines functionalized by acylation and azide-alkyne ligation reactions. In addition, VLPs such as Qβ impart strong immunogenicity to attached molecules including carbohydrates [79,80] by virtue of their regular structure and self-adjuvanting properties [81].

We explored acute Chagas disease in infected α-GalT-KO mice at 8 days post-infection (dpi) with the Y strain and 22 dpi for the Colombian strain, whereas the chronic stage was evaluated at 36 and 190 dpi, respectively. The Qβ-αGal particle was highly effective, producing high titers of anti-αGal IgG antibodies, controlling parasitemia, enhancing survival upon challenge with a very high concentration of *T. cruzi* in strains belonging to different DTUs (I and II). It is also crucial for patients infected with *T. cruzi* to prevent cardiac damage by long-term exposure to amastigote colonies established in the heart tissue. Therapeutic vaccination has previously demonstrated a correlation between the reduction of parasitemia and easing of long-term cardiac parasite burden [15,82–84]. In our case, all animals vaccinated and infected with two different *T. cruzi* parasite strains did quite well initially and also cleared the parasites from the heart at 36 dpi for Y strain and 190 dpi for Colombian strain. This led to significant improvements in heart inflammation, hypertrophy, lesion intensity, and degenerative changes in cardiac tissues. The Qβ-αGal particle was found to function as a TL4 agonist *in vitro* experiments, inducing macrophages to produce more nitric acid. Correspondingly, higher proinflammatory cytokines IL-12 and IFN-γ were found in heart homogenates of vaccinated animals. The result was a promising combination of protection against primary infection and an ability to clear amastigote parasites from the heart, thereby limiting the damage by acute and chronic Chagas disease infection. The results showed here suggest that Qβ-αGal particles could be a candidate for vaccinating different parasite DTUs.

## Material and methods

### Ethics statement

All experiments were approved and conducted according to the guidelines of the Ethics Committee on the Use of Animals (CEUA) of the Federal University of Minas Gerais (protocol n˚ 255/2013).

**Mice.** Mice (*Mus musculus*), females aged 6 to 8 weeks of the C57BL/6 lineage, depleted the gene of α1,3-galactosyltransferase enzyme (αGalT-KO), were used. The mice were donated by Director Peter Cowan, Hospital São Vicente, Australia and by Dr. Kim Janda, Scripps Research Institute, United States, and are bred and maintained in the vivarium of the Federal University of Minas Gerais, Department of Parasitology. Under appropriate conditions of temperature and humidity, the mice were housed in 30.3 x 19.3 x 12.6 cm polypropylene boxes with 3–5 mice each, with controlled 12-hour light-dark cycles, receiving commercial feed specific to the species (Presence/Archer Daniels Midland Company) and water *ad libitum*.

**Parasites.** Trypomastigote forms of the Y and Colombian strains of *T. cruzi* were kept in the *T. cruzi* Biology Laboratory, Federal University of Minas Gerais. The Y strain of *T. cruzi*—DTU TcII ([77], Zingales et al., 2009) was isolated from a patient in the acute phase of Chagas disease by Pereira de Freitas, in 1950, in Marília, São Paulo and later studied and described by Silva and Nussenzweig [85]. The Colombian strain of *T. cruzi*—DTU TcI [77] was isolated from a patient in Colombia and later studied and described by Federici, *et al* [86]. Parasites were kept in mice and recovered for challenges.

**Parasitemia.** The parasitic load was estimated from the parasite count in 5 μL blood samples collected from the caudal vein of the mouse, made under an optical microscope at 40X using the Brener method. [87] The number of mobile blood trypomastigotes was counted in 50 fields at random distributed throughout the slide area. The data were demonstrated as the number of trypomastigotes per mL of blood [87].

**ELISA.** Blood samples were obtained by submandibular collection before the first immunization and five days after the mice received the final (first or second) dose. Blood samples were also collected by cardiac puncture at the time of euthanasia. All samples were stored in tubes without anticoagulant. Serum aliquots were obtained by centrifugation at 3000 x g for 15 minutes at 25°C and stored at -20°C until the time of use. To perform the immunoenzymatic assay, polystyrene microplates with 96 wells (NUNC MaxiSorp/Thermo Fisher Scientific) were treated with 50 μL of the selected particle (Qβ-αGal, Qβ-WT) at 0.4 μg/mL in 100 mM carbonate/sodium bicarbonate buffer, pH 9.6, for 18 hours at 2–8°C. The wells were then blocked with 200 μL of PBS1X solution containing 1% bovine serum albumin (PBS1X - BSA 1%), for 50 minutes at 37°C. After blocking, 50 μL of diluted serum (1:100) were added in triplicate in PBS1X - BSA 1% solution, and the microplates were incubated for 90 minutes at 37°C. Wells were then washed three times with 200 μL of PBS1X containing 0.05% Tween-20 (PBS1X –T 0.05%) per well. Anti-mouse IgG antibody conjugated with biotin (GE Healthcare) diluted 1:5,000 in PBS1X - BSA 1% solution (50 μL) was added, and the microplates were incubated for 50 minutes at 37°C. After washing three times, 50 μL of streptavidin-conjugated horseradish peroxidase (GE Healthcare) diluted 1:3,000 in 1% PBS1X - BSA solution was added and incubated for 50 minutes at 37°C. The plate was washed three times, each well was treated with substrate solution (100 μL), and incubated for 30 minutes in the dark, followed by reading on a microplate reader at 490 nm. The substrate solution consisted of 2 mg OPD (o-phenylenediamine dihydrochloride) and 4 μL of 30–32% hydrogen peroxide in 10 mL of a solution containing 30 mM citric acid, 50 mM disodium phosphate (pH 5), diluted in 1 mL of distilled water.

**Cytokines.** Immediately after collection from euthanized mice, a sample (~1g) of cardiac tissue was subjected to maceration in 1 mL of extraction buffer containing protease inhibitor. The resulting homogenate was centrifuged for 10 minutes at 10,000 rpm at 4°C, and the supernatant was collected and stored at -80°C for cytokine determination using the BD OptEIA Set Mouse kits (BD Biosciences) to determine IL-12 p40 and IFN-γ levels using the capture ELISA assay. The observed detection limits were 15.6 pg/mL for IL-12p40 and 3.1 pg/mL for IFN-γ.

**Histopathological analysis of inflammation, degenerative changes, hypertrophy, and amastigote nests.** Groups of mice, six per group, were euthanized, a fragment of each excised

heart was fixed in 10% buffered formaldehyde for 7 days. Then it was dehydrated in increasing alcoholic dilutions, diaphanized in xylol, infiltrated and included in paraffin for block making and subsequent microtomy, gluing, and mounting the blades. The blocks were cut into sections of 4 μm in diameter for staining with hematoxylin and eosin (H&E). After making the slides, quantitative and semi-quantitative histopathological analyses were performed by optical microscopy. The lesions displayed in the myocardium were assessed regarding inflammatory infiltrate, degenerative changes, and hypertrophy. The presence of amastigote nests quantified parasitism in cardiac tissues. For semi-quantitative analyzes, the slides were examined in a bright field optical microscope coupled with a digital image capture system (Motic 2.0). For the score of myocardial inflammation, ten random images were captured per fragment with a 20X magnification. The rating was based on four degrees of myocardial inflammation: grade 0 (absent) = absence of inflammatory cells around the cardiomyocytes; grade 1 (discrete) = some cardiomyocytes had a small number of inflammatory cells; grade 2 (moderate) = some cardiomyocytes had significant inflammation; grade 3 (marked) = some cardiomyocytes had an intense inflammatory infiltrate. A four-point scoring system was also adopted for degenerative changes represented by tissue necrosis, autolysis, or cardiomyocyte degeneration: grade 0 (absent) = absence of histopathological changes; grade 1 (slight) = some cardiomyocytes showed slight degeneration; grade 2 (moderate) = some cardiomyocytes had a degenerative aspect, and others were in autolysis; grade 3 (marked) = numerous cardiomyocytes had a degenerative aspect, autolysis. For the semi-quantitative analysis of cardiomyocyte hypertrophy, a four-point intensity scale was used: grade 0 (absent) = no cardiomyocyte hypertrophy observed; grade 1 (discreet) = few cardiomyocytes were hypertrophied; grade 2 (moderate) = a significant number of cardiomyocytes were observed to have hypertrophy; grade 3 (marked) = large numbers of cardiomyocytes were hypertrophied. To count the amastigote nests present in the histological cuts in the cardiac muscle tissue, a complete scan of the heart sections of the mice was performed with a 20X magnification.

**TLR2** [(-/-)] **and TLR4** [(-/-)] **murine peritoneal macrophages.** **TLR2[(-/-)] and TLR4[(-/-)] knockout strains of C57BL/6 mice were obtained from Oswaldo Cruz Institute, Fiocruz. Thioglycolate (2 mL) was injected in the mouse peritoneum and macrophages were extracted by peritoneal washing with ice-cold RPMI, and enriched by plastic adherence (1 h, 37˚C, 5% $CO_2$). Cells (3 x $10^5$ cells/well) were washed with fresh RPMI then cultured in RPMI containing 2 mM glutamine, 50 U/mL of penicillin, and 50 μg/mL streptomycin supplemented with 10% FBS in 96-well culture plates (37˚C, 5% $CO_2$). Cells were primed with interferon-gamma (IFN-γ, 3 IU/mL) for 18 h before incubation with LPS, Qβ-gal, Qβ-control (10 μg/mL), or only medium. Nitrite concentrations were determinate by the Griess reaction (Griess Reagent System, 2009).

**Statistical analyses.** GraphPad Prism software (San Diego, USA, version 7.0) was used for statistical analysis. The bidirectional analysis of variance test (ANOVA), followed by the Bonferroni post-test, was used to compare the values presented by the parasitemia and weight variation curves. The Log Rank test was used to compare the survival rate of mice after infection with *T. cruzi*. Fisher's test was used to compare the different proportions of mice that survived the infection. Statistical analyses were also used to compare maximum levels of parasites detected in peripheral blood, dosages of total IgG antibodies, the intensity of inflammation, degenerative changes, hypertrophy, parasitic load, and cytokine profile in cardiac tissues. For this, the Student's T-test parametric data and the Mann Whitney test for non-parametric data were used in comparative analyses between two groups. For the comparative analysis between three or more groups, the one-way ANOVA test was used, followed by the Tukey post-test, for parametric data, and the Kruskal-Wallis test, followed by the Dunns post-test for non-parametric data. To check whether the data presented Gaussian distribution, the D'Agostino &

Pearson normality test was used. Differences between groups were considered significant when the p-value <0.05.

## Author Contributions

**Conceptualization:** M. G. Finn, Alexandre Ferreira Marques.

**Data curation:** Fabricio Marcus Silva Oliveira, M. G. Finn, Alexandre Ferreira Marques.

**Formal analysis:** Gisele Macêdo Rodrigues da Cunha, Maíra Araújo Azevedo, Edward Valencia Ayala, Juan Atilio Jimenez Chunga, Raul Jesus Ynocente La Valle, Fabricio Marcus Silva Oliveira.

**Funding acquisition:** Ricardo Gazzinelli, M. G. Finn, Alexandre Ferreira Marques.

**Investigation:** Gisele Macêdo Rodrigues da Cunha, Maíra Araújo Azevedo, Denise Silva Nogueira, Marianna de Carvalho Clímaco, Edward Valencia Ayala, Juan Atilio Jimenez Chunga, Raul Jesus Ynocente La Valle, Fabricio Marcus Silva Oliveira.

**Methodology:** Gisele Macêdo Rodrigues da Cunha, Maíra Araújo Azevedo, Lucia Maria da Cunha Galvão, Egler Chiari, Carlos Ramon Nascimento Brito, Rodrigo Pedro Soares, Paula Monalisa Nogueira, Ricardo Toshio Fujiwara, Ricardo Gazzinelli, Robert Hincapie, Carlos-Sanhueza Chaves, Fabricio Marcus Silva Oliveira.

**Writing – review & editing:** M. G. Finn, Alexandre Ferreira Marques.

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
