## [Decision Letter · Decision Letter 0]

3 May 2021

Dear Dr. Marques,

Thank you very much for submitting your manuscript "α-Gal Immunization Positively Impacts Trypanosoma cruzi Colonization of Heart Tissue in a Mouse Model" for consideration at PLOS Neglected Tropical Diseases. As with all papers reviewed by the journal, your manuscript was reviewed by members of the editorial board and by several independent reviewers. In light of the reviews (below this email), we would like to invite the resubmission of a significantly-revised version that takes into account the reviewers' comments. 

Dear Colleagues,

Reviewer #2 raised significant concerns and required more experiments in addition to modifications on some figures and the text. The necessary experiments are of capital importance to strengthen the main concepts underlined in the manuscript. 

The authors must address the points raised by both reviewers and the experiments suggested by the reviewer.

We cannot make any decision about publication until we have seen the revised manuscript and your response to the reviewers' comments. Your revised manuscript is also likely to be sent to reviewers for further evaluation.

Sincerely,

Ulisses Gazos Lopes

Associate Editor

Walderez Dutra

Deputy Editor

Dear Autho

Reviewer #2 raised significant concerns and required more experiments in addition to modifications on some figures and the text. The necessary experiments are of capital importance to strengthen the main concepts underlined in the manuscript. 

The authors must address the points raised by both reviewers and the experiments suggested by the reviewer.

Reviewer's Responses to Questions

**Key Review Criteria Required for Acceptance?**

**Methods**

-Are the objectives of the study clearly articulated with a clear testable hypothesis stated?

-Is the study design appropriate to address the stated objectives?

-Is the population clearly described and appropriate for the hypothesis being tested?

-Is the sample size sufficient to ensure adequate power to address the hypothesis being tested?

-Were correct statistical analysis used to support conclusions?

-Are there concerns about ethical or regulatory requirements being met?

Reviewer #1: The objectives as well as the hypothesis are clear.

The proposed methodology is adequate to meet the established objectives.

The population is clearly described and appropriate.

The sample size is sufficient.

The statistical analysis is correct.

The study adheres to the guidelines of the Ethics Committee.

Reviewer #2: -Are the objectives of the study clearly articulated with a clear testable hypothesis stated? YES

-Is the study design appropriate to address the stated objectives? In most cases, yes, it is. However, when testing the hypothesis that the Qb-aGal VLP is a TLR4 agonist, further experiments are necessary.

-Is the population clearly described and appropriate for the hypothesis being tested? YES

-Is the sample size sufficient to ensure adequate power to address the hypothesis being tested? 

In most experiments, yes, it is. However, for the results shown in Figs. 3D and 3F, no statistical significance is attained when comparing Qb-aGal- and Qb-WT-treated groups. Of notice, there is no information about what errors bars represent in any figure: standard deviation or standard error of the mean? This information should be available in every Figure caption. I suggest showing the standard error of the mean in every graph. Concerns about the interpretation of results shown in Figs. 3D and 3F will be further commented below. Also, sample size is not specified in Figs.4B-4G caption.

-Were correct statistical analysis used to support conclusions? YES.

-Are there concerns about ethical or regulatory requirements being met? NO

**Results**

-Does the analysis presented match the analysis plan?

-Are the results clearly and completely presented?

-Are the figures (Tables, Images) of sufficient quality for clarity?

Reviewer #1: The analysis matches what was planned.

The results are clearly presented.

The figures are sufficient and of very good quality.

Reviewer #2: -Does the analysis presented match the analysis plan? YES

-Are the results clearly and completely presented? YES

-Are the figures (Tables, Images) of sufficient quality for clarity? Yes, with the exception of figures showing parasitemia kinetics (Figs. 1C, 2A and 3B), in which red dotted lines are employed to show the upper limit of the standard deviation for each set of data. This is very confusing. In my opinion, error bars (showing the standard error of the mean at each point - day pi) should be plotted instead, in each of the 3 parasitemia curves (corresponding to the 3 groups: Qb-aGal, Qb-WT and PBS).

**Conclusions**

-Are the conclusions supported by the data presented?

-Are the limitations of analysis clearly described?

-Do the authors discuss how these data can be helpful to advance our understanding of the topic under study?

-Is public health relevance addressed?

Reviewer #1: The conclusions are sufficiently supported as they are supported by the data presented.

The limitations of the analysis are not clearly described.

Discussion of the data is useful to understand its scope and impact in the area.

If the relevance to public health is addressed.

Reviewer #2: -Are the conclusions supported by the data presented? 

In most cases, yes, they are. However, when presenting the results of Figs. 3D and 3F, on page 6, the authors state: "Although not statistically significant, we observed decreased hypertrophy (Fig. 3E) and lesion intensity in the myocardium of the vaccinated group compared with the animals receiving the underivatized particle". If no statistical significance was attained when comparing Qb-aGal- and Qb-WT-treated groups, it cannot be stated that there was a decrease in hypertrophy and lesion intensity. The authors should either increase the sample sizes, in order to attain statistical significance, or rephrase the statement. It is not possible to conclude that there is decreased hypertrophy and lesion intensity, without attaining statistical significance.

Also, the conclusion that IL-12 and IFN-g are produced in response to Qb-aGal through the activation of TLR4 in macrophages of the heart is not supported by presented data (please see details in point #5, below).

-Are the limitations of analysis clearly described? YES

-Do the authors discuss how these data can be helpful to advance our understanding of the topic under study? YES

-Is public health relevance addressed? YES

**Editorial and Data Presentation Modifications?**

Reviewer #1: There are no editorial suggestions.

Reviewer #2: 1. All over the Introduction and Discussion sections, a period punctuation mark is wrongly placed before the indicated reference numbers.

2. On the last line of page 3: ..." infection by T. cruzi Y and Colombian strains who displays different levels of ..." correct: ...strains, which display different...

3. Page 6: "... we observed decreased hypertrophy (Fig. 3E) and lesion intensity..." Correct: (Fig. 3D).

The manuscript lacks page and line numbers: this does not help at all the work of a reviewer. Should be required for every submitted ms.

**Summary and General Comments**

Reviewer #1: Summary.

In this study the effectiveness of a vaccine against Chagas disease was tested. alfa-GalT-KO mice were immunized twice with a virus-like particle vaccine displaying a high density of the immunogenic alfa-Gal trisaccharide (Qbeta-αGal). Seven days after the second immunization the animals were infected with trypomastigotes of two different strains of T. cruzi.

They analyzed the effect of protection at the humoral and histological level. They concluded that this vaccine can offer protection in case of infection with different DTUs of T. cruzi, reducing its effect in the acute phase and avoiding damage to the heart in the chronic phase.

General Comments.

1. Although the authors mention two references (I can only access one reference), it is highly necessary that in the methodology of this manuscript they briefly describe the vaccine preparation process and emphasize the composition of the vaccine. 

2.-The authors name an experimental group in three different ways and it causes confusion. They are required to unify the nomenclature or specify what each of the groups corresponds to:

QB-control

QB-WT

QB-VLPs

3.-In the first paragraph of results, figure 1 is mis-referenced:

where the authors say Fig. 1A they should say Fig. 1B

where the authors say Fig. 1B they should say Fig. 1C

where the authors say Fig. 1C they should say Fig. 1D

At the end of the statement "To test protection ... QB-VLPs or PBS" the authors should refer to Fig. 1A.

4.- Also in the second paragraph of results, figure 1 is mis-referenced:

where the authors say Fig. 1A they should say Fig. 1B.

5.-How do you interpret the similar hypertrophy result between the QB-Glu and QB-alpha Gal groups?

Reviewer #2: In the present manuscript, Gisele M. R. da Cunha et al., assess Qb-aGal (virus-like particles [VLP] functionalized with a dense array of synthetic a-Gal trisaccharide) as a candidate for vaccine development against infection with T. cruzi. For this, they immunized and challenged α1,3-galactosyltransferase knockout mice. The same Gal (Gal-a1,3-Gal-β1,4-GlcNAc) motif was previously employed covalently linked to a carrier protein (HSA), and tested as an experimental vaccine against infection with T. cruzi [ref. 30]. On the other hand, the same Qb-aGal VLPs were employed as an experimental vaccine against infection with other trypanosomatids: Leishmania amazonensis and L. infantum, in a1,3-galactosyltransferase knockout mice, by the same group [ref. 27].

 Interesting results are shown: similarly to what was previously shown with the a-Gal-HSA vaccine, immunization with Qb-αGal reduced parasitemia and mortality, as well as inflammation score and degenerative changes in the heart tissue of mice challenged with 2 strains of T. cruzi, belonging to different DTUs (I and II). Qb-WT VLPs were employed as control. The advantages of the present vaccine upon the previous aGal-HSA (ref#30) should be addressed in more details.

 Although the majority of the experiments are well conducted and controlled, some points should be clarified and certain statements rephrased. Also, some missing experimental controls and information should be added, as specified below.

1) When presenting the results of Figs. 3D and 3F, on page 6, the authors state: "Although not statistically significant, we observed decreased hypertrophy (Fig. 3D - not 3E) and lesion intensity in the myocardium of the vaccinated group compared with the animals receiving the underivatized particle". If no statistical significance was attained when comparing Qb-aGal- and Qb-WT-treated groups, it cannot be stated that there was a decrease in hypertrophy and lesion intensity. The authors should either increase the sample sizes, in order to attain statistical significance, or rephrase the statement. It is not possible to conclude that there is decreased hypertrophy and lesion intensity, without attaining statistical significance.

2) Regarding hypertrophy, in the Material and Methods section (page 12, last line) it is said that a semi-quantitative analysis of cardiomyocyte hypertrophy was performed. This is different from heart hypertrophy, a characteristic of the chronic cardiac form of Chagas disease. Therefore, to avoid misunderstanding, all through the text the authors should refer to "cardiomyocyte hypertrophy" and not to "heart hypertrophy" or simply hypertrophy, as in pages 5, 6 and 10.

3) In the Discussion section (3rd paragraph on page 9), it is stated that a previous work has shown that "...both CD4+ and CD8+ T cells were activated by immunization, protecting a-GalT-KO mice by decreasing parasitemia and the number of parasites found on heart tissues.[30]". However, in fact, this was NOT demonstrated in ref. #30. This previous work showed an experiment where total splenocytes from vaccinated and non-vaccinated mice (challenged or not) were incubated in vitro for 24 h in the presence of aGal-HSA and the frequency of CD4+ and CD8+ T cells was measured by flow cytometry afterwards. Frequency of CD44+ T cells was also evaluated. Absolute cell numbers were not shown. This experiment is not a proof of Ag-specific T cell activation. Certain cell-subset percentages can be increasing simply because other splenocytes might be dying in vitro, for example. Also, in vitro incubation for only 24 h does not induce T cell proliferation. Of course, since IgG anti-aGal, are induced by vaccination, CD4+ T cells are in all likelihood being activated too, but probably this is the consequence of the recognition of MHC II-restricted peptides derived from the carrier ptn. In challenged mice, CD44+ T cells were probably activated by parasite-derived peptides. No test on the Ag-specificity of T cells were performed in ref #30, nor in the present study. Importantly, no evidence is shown that T cells are responsible for "decreasing parasitemia and the number of parasites found on heart tissues in vaccinated mice as stated on page 9 of the present manuscript. Therefore this affirmation must be removed.

4) Fig. 4A shows evidence that Qb-aGal could be acting as a TLR4 agonist. Peritoneal macrophages from WT and Tlr2-/- or Tlr4-/- mice were stimulated in vitro with Qb-aGal or Qb-WT VLPs, but only nitric oxide production was measured. TLR4 agonists usually also induce cytokine production, such as IL-6, TNF and type I IFN. The demonstration that Qb-aGal also induces these cytokines in WT but not in Tlr4-/- macrophages would strengthen this evidence. It is also very important to demonstrate that Qb-aGal VLPs are not contaminated with endotoxin, by performing the LAL assay, for example. 

5) Also concerning Fig.4, its title is not appropriate, since TLR activity was tested only in vitro and with cells from non-vaccinated mice. The fact that the heart homogenates of mice vaccinated with Qb-aGal produce more IL-12 and IFN-g (Fig. 4B-4E) is not an evidence of TLR4 activation at all, since other pathways can also induce these cytokines. Furthermore, since these animals were not only vaccinated, but also infected, production of IL-12 and IFN-g might be driven by the parasite and not by the vaccine. Corroborating with this idea is the fact that IL-12 and IFN-g measurement was performed several days post-infection (8, 22, 36 and 190 dpi), at time points in which Qb-aGal was most probably already eliminated from the vaccinated mice. In the Discussion section it is stated that: "The Qb-aGal particle was found to function as a TLR4 agonist, inducing macrophages to produce more nitric acid, IL-12, and TNF-g in heart tissues". However, no evidence was shown that macrophages are the cells producing these cytokines in the heart, or that the production of these cytokines is dependent on TLR4 activation. Therefore, the text referring to these results (on page 7) should be rephrased. Also, the inaccurate conclusion that these results are an evidence of TLR4 activation in heart macrophages by vaccination with Qb-aGal should be removed from the last paragraph of the Discussion section, on page 10.

Minor: 

1) I suppose Fig. 4F shows results of mice infected with Y strain, while Fig. 4G with Colombian strain. This information should be added to the figure caption.

2) In the Material and Methods section, a description of how Q�-aGal VLPs are obtained is completely absent and must be included.

PLOS authors have the option to publish the peer review history of their article (what does this mean?). If published, this will include your full peer review and any attached files.

Reviewer #1: No

Reviewer #2: No
---

## [Editor Report · Decision Letter 1]

30 Jun 2021

Dear Dr. Marques,

We are pleased to inform you that your manuscript 'α-Gal Immunization Positively Impacts Trypanosoma cruzi Colonization of Heart Tissue in a Mouse Model' has been provisionally accepted for publication in PLOS Neglected Tropical Diseases.

Best regards,

Ulisses Gazos Lopes

Associate Editor

Walderez Dutra

Deputy Editor

---

## [Editor Report · Acceptance letter]

23 Jul 2021

Dear Dr. Marques,

We are delighted to inform you that your manuscript, "α-Gal Immunization Positively Impacts Trypanosoma cruzi Colonization of Heart Tissue in a Mouse Model," has been formally accepted for publication in PLOS Neglected Tropical Diseases.

Best regards,

Shaden Kamhawi

co-Editor-in-Chief

Paul Brindley

co-Editor-in-Chief
